# From Chronodisruption to Sarcopenia: The Therapeutic Potential of Melatonin

**DOI:** 10.3390/biom13121779

**Published:** 2023-12-12

**Authors:** José Fernández-Martínez, Yolanda Ramírez-Casas, Yang Yang, Paula Aranda-Martínez, Laura Martínez-Ruiz, Germaine Escames, Darío Acuña-Castroviejo

**Affiliations:** 1Centro de Investigación Biomédica, Facultad de Medicina, Departamento de Fisiología, Instituto de Biotecnología, Parque Tecnológico de Ciencias de la Salud, Universidad de Granada, 18016 Granada, Spain; josefermar@ugr.es (J.F.-M.); yolandaramirez@correo.ugr.es (Y.R.-C.); ampaula@correo.ugr.es (P.A.-M.); lauramartinezruiz8@gmail.com (L.M.-R.); gescames@ugr.es (G.E.); 2Instituto de Investigación Biosanitaria (Ibs.Granada), Hospital Universitario San Cecilio, 18016 Granada, Spain; 3Key Laboratory of Resource Biology and Biotechnology in Western China, Ministry of Education, Faculty of Life Sciences and Medicine, Northwest University, Xi’an 710069, China; yang200214yy@163.com; 4Centro de Investigación Biomédica en Red de Fragilidad y Envejecimiento Saludable (CIBERFES), Instituto de Salud Carlos III (ISCIII), 28029 Madrid, Spain; 5UGC de Laboratorios Clínicos, Hospital Universitario San Cecilio, 18016 Granada, Spain

**Keywords:** aging, chronodisruption, clock genes, *Bmal1*, inflammaging, oxidative stress, mitochondria, skeletal muscle, sarcopenia, melatonin

## Abstract

Sarcopenia is an age-related condition that involves a progressive decline in muscle mass and function, leading to increased risk of falls, frailty, and mortality. Although the exact mechanisms are not fully understood, aging-related processes like inflammation, oxidative stress, reduced mitochondrial capacity, and cell apoptosis contribute to this decline. Disruption of the circadian system with age may initiate these pathways in skeletal muscle, preceding the onset of sarcopenia. At present, there is no pharmacological treatment for sarcopenia, only resistance exercise and proper nutrition may delay its onset. Melatonin, derived from tryptophan, emerges as an exceptional candidate for treating sarcopenia due to its chronobiotic, antioxidant, and anti-inflammatory properties. Its impact on mitochondria and organelle, where it is synthesized and crucial in aging skeletal muscle, further highlights its potential. In this review, we discuss the influence of clock genes in muscular aging, with special reference to peripheral clock genes in the skeletal muscle, as well as their relationship with melatonin, which is proposed as a potential therapy against sarcopenia.

## 1. Sarcopenia: Age-Related Loss of Muscle Mass and Function

### 1.1. Aging

Aging can be broadly defined as the time-dependent functional decline that affects most living organisms. It is characterized by a gradual loss of physiological integrity, leading to diminished function and increased susceptibility to mortality. This deterioration is the primary risk factor for the development of significant human pathologies, including cancer, diabetes, cardiovascular disorders, neurodegenerative diseases, and sarcopenia, among others. Fortunately, research on aging has experienced an unprecedented advancement in recent years, especially with the discovery that the rate of aging is controlled, at least to some extent, by genetic pathways and biochemical processes conserved during evolution [1,2].

Clearly, the aging of the population, along with the emergence of age-related diseases and other associated challenges, is becoming a primary concern for humanity. Every country in the world is experiencing growth in both the number and percentage of elderly individuals in the population. It is projected that between 2022 and 2050, the proportion of the global population aged 65 years or over will increase from 10% to 16%. By 2050, the number of individuals aged 65 or older will be more than double the number of children under 5 years old and roughly equal to the number of children under 12 years old (United Nations, DESA, Population Division. World Population Prospects 2022. http://population.un.org/wpp/ (accessed on 8 October 2023)).

This demographic shift, as a direct result of lower mortality and increased survival to advanced ages, coupled with a sustained decrease in fertility levels (United Nations, DESA, Population Division. World Population Prospects 2022. http://population.un.org/wpp/ (accessed on 8 October 2023)), is accompanied by another transition, the epidemiological transition, characterized by shifts in prevalent diseases [3]. Consequently, disability prevention and the preservation of health and independence in older individuals are now among the primary goals of healthcare. In this context, sarcopenia is an especially serious issue as people live longer, as it is a condition where skeletal muscle deficiency leads to loss of functionality and fragility, further impairing the quality of life for those affected by it.

### 1.2. Definition and Primary Characteristics of Sarcopenia

Sarcopenia is a term derived from the Greek *σάρξ* (sarx, meaning “flesh”), and *πενία* (penia, meaning “poverty” or “scarcity”). It was first described in the 1980s as an age-related decrease in body muscle mass that affects mobility, nutritional status, and independence [4]. The definition has evolved since then, marked by two recent milestones: the inclusion of muscle function (defined by muscle strength, power, and physical performance) in the concept, as it has been shown to be a clinically more powerful predictor than muscle mass alone; and the recognition of sarcopenia as an independent condition with a code in the International Classification of Diseases-10 (ICD-10) [5,6]. However, most clinicians are still unaware of the condition and the necessary diagnostic tools to identify it. 

Even today, the most cited definition for sarcopenia was proposed by the European Working Group on Sarcopenia in Older People (EWGSOP), endorsed by the Asian Working Group on Sarcopenia (AWGS), and updated as EWGSOP2 in January 2019. In this definition, sarcopenia is described as a progressive and widespread disorder of skeletal muscle involving accelerated loss of muscle mass and function, associated with an increase in adverse outcomes such as falls, functional decline, frailty, and mortality [7].

Sarcopenia is a universal phenomenon with a complex and multifactorial etiology. It commonly occurs as an age-related process in older individuals, although it can also occur in middle age in association with a variety of conditions. Factors involved in sarcopenia include aging as a primary cause, but also genetic factors, inactivity or a sedentary lifestyle, nutritional causes (low protein and energy intake, vitamin D deficiency, …), bone and joint diseases, cardiorespiratory disorders, hormonal changes (decrease in testosterone and growth hormone levels), diabetes, neurological diseases, cancer, liver and kidney disorders, and iatrogenic factors (hospitalization, drug-related side effects) [5,8].

In clinical practice, it is established that an individual with low muscle strength and low muscle mass or quality should be diagnosed with sarcopenia. The condition can be better understood as a failure or insufficiency of skeletal muscle [9]. Thus, sarcopenia can manifest acutely (usually in the context of an acute illness or sudden immobility, such as during a hospitalization) or have a more prolonged (chronic) course [5]. Muscle mass and strength (along with bone mineral density) peak in young adulthood and, after a period of stability, begin to gradually decline with age, with a faster decline in strength [10]. However, most cases of sarcopenia go undiagnosed, only coming to attention when relevant symptoms are reported. These symptoms could include falls, weakness, slowness, self-reported loss of muscle mass, or difficulties in performing daily activities. Detecting cases is particularly relevant in care settings where a higher prevalence of sarcopenia might be expected, such as temporary hospital admissions, rehabilitation environments, or nursing homes [11]. 

Indeed, the detection of sarcopenia is not so straightforward. At times, sarcopenia may be associated with thinness, and it may not be recognized that it can also be present in obesity. Sarcopenic obesity is often identified when a person exhibits both low muscle mass and increased adiposity, but it might go unnoticed when the focus of attention is on obesity, leading to adverse outcomes [12]. Sarcopenia can also be mistaken for other conditions like malnutrition or cachexia, although it is true that it can coexist with both. These conditions involve a reduction in weight and muscle mass, but only sarcopenia entails a decrease in muscle strength with impaired muscle function. Sarcopenia is also closely linked to physical frailty, described as the biological substrate of the frailty phenotype, which involves involuntary weight loss, self-reported exhaustion, weakness (low grip strength), slow walking speed, and low physical activity [13,14]. As we can see, sarcopenia can occur in association with a variety of chronic conditions in middle age, with most patients presenting more than one associated condition. 

Research in this area has been challenging for several reasons, including differing opinions on the definition of sarcopenia, the increasing recognition of acute and chronic forms of sarcopenia, the existence of many interactive pathways involved in its pathophysiology, and the impact of related conditions (including those that might mimic sarcopenic symptoms and other existing patient conditions that affect sarcopenia).

### 1.3. Sarcopenia Management

The treatment of sarcopenia encompasses both non-pharmacological and pharmacological options. Resistance exercise and proper nutrition are crucial [15]. Evidence supports that resistance exercise improves muscle mass, strength, and physical performance [16,17,18,19,20], while the effectiveness of nutritional intervention alone is not entirely clear, though certain dietary patterns, such as adequate protein intake, vitamin D, antioxidant nutrients, and long-chain polyunsaturated fatty acids have shown benefits [21]. Nevertheless, these methods may not suffice to prevent or reverse this disease, especially to counteract the numerous pathophysiological mechanisms leading to sarcopenia (see below). Furthermore, in severely debilitated patients implementing exercise-based treatments may be impractical. The Food and Drug Administration (FDA) has not approved specific drugs for sarcopenia. However, various agents are recommended, including growth hormones, anabolic steroids or androgens like testosterone, protein anabolic agents, appetite stimulants like ghrelin, myostatin inhibitors, among others, although their effectiveness and side effects vary [22].

### 1.4. Pathophysiology of Sarcopenia

While behaviors like physical inactivity or poor nutrition are known to contribute to age-related loss of muscle mass and function, they are not the sole explanation [23]. In fact, loss of muscle mass and function has been reported in veteran athletes who maintain a good diet [24,25,26], suggesting that other factors influence this debilitating condition. Currently, the specific mechanisms underlying sarcopenia are not clear, although some processes implicated in this pathology have been proposed.

Aging disrupts the homeostasis of skeletal muscle, resulting in an imbalance of muscle proteins and leading to a general loss of skeletal muscle mass. Additionally, it causes the loss of other critical components of myocytes, including organelles such as mitochondria, as well as various proteins from skeletal muscle [3,23]. Many of these gene products are involved in processes regulated by circadian control, such as genes implicated in mitochondrial health and metabolism (*Pgc1*, *Pink*, *Parkin*, *Bnip3*), myogenesis (*Myod1*, *p300*), myokine secretion (*IL-6*, *IL-8*), and protein synthesis and degradation (*4ebp1*, *Atrogin*, *Murf1*, *Ubc*). Several of these components play a crucial role in the regulation of muscle mass [27]. In general, this translates to a decrease in both the size and number of muscle fibers [3,23]. From ages 20 to 80, there is approximately a 30% reduction in muscle mass and a decrease in the cross-sectional area of around 20% [28]. This is particularly significant in the lower limbs, significantly impacting mobility [3]. The skeletal muscle of most mammalian species, including rats and mice, is composed of four types of fiber populations: slow type 1 (oxidative), fast type 2A (oxidative), fast type 2B (glycolytic), and fast type 2X (glycolytic). However, only type 1, 2A, and 2X fibers are present in human muscles. In both humans and other mammalian species, we also find hybrid fibers, formed by different combinations of the various fiber types [29,30]. During aging, morphological changes occur in muscle fibers, such as an increase in the variability of fiber size, accumulation of ungrouped, scattered, and angular fibers, nuclei located in central positions, expansion of the extracellular space, and deposition of protein aggregates within the interstitial matrix. These morphological changes occur alongside increased infiltration of non-contractile tissue, such as adipose tissue (myosteatosis) or connective tissue (fibrosis). The changes in sarcopenic muscle particularly affect type 2 fibers, which are crucial for activities involving strength and/or power. With age, there is a transition of muscle fibers from type 2 to type 1; along with a decrease in the number of satellite cells (MuSCs) and their proliferative and regenerative capacity, especially in type 2 fibers; an increase in hybrid fibers; and loss of motor neurons [3,5,8,23,31]. These changes contribute to declining muscle functional capacity, which in turn contributes to functional disability. In fact, several studies have reported that loss of muscle mass may not be the sole explanation for the loss of muscle strength and physical function in older adults [3]. In addition, several mechanisms associated with aging contribute to this loss of muscle mass and functional decline, including age-dependent inflammation, oxidative stress, loss of mitochondrial respiratory capacity, and apoptosis, as well as changes in hormones (mainly sex hormones and thyroid hormones) and growth factors, and microvascular changes [3,5,23].

### 1.5. Connection between Inflammation, Oxidative Stress, Mitochondrial Dysfunction, and Apoptosis in Aging

Normal aging is accompanied by a chronic low-grade inflammation, named “inflammaging” [32], which is accompanied by oxidative stress and mitochondrial impairment, and is directly related to frailty [33]. During aging, there is an increase in the activation of the nuclear factor κB (NF-κB) inflammation pathway promoted, among other factors, by increased oxidative stress, either through greater production of free radicals (reactive nitrogen species (RNS) and primarily reactive oxygen species (ROS)), or reduced capacity of antioxidant defense, or both [34,35]. Although the term “mitohormesis” was proposed to show that increased ROS can cause a mild oxidative stress that in turn promotes mitochondrial defenses, this only occurs with a low level of ROS [36]. The role of free radicals in aging is a much more complex process. Free radicals increase with age and induce the activation of NF-κB and its translocation to the nucleus, where it controls the expression of nearly 200 genes, including multiple genes leading to inflammatory proteins and proinflammatory cytokines, including pro-IL-1β. This situation creates a prooxidant environment that can damage mitochondria, the primary source of ROS formation, to such an extent that it leads to an overproduction of free radicals [35]. This mitochondrial oxidative stress, in turn, causes the opening of the mitochondrial permeability transition pore (MPTP), releasing free radicals and mitochondrial DNA into the cytosol [35,37]. These molecules trigger the NLRP3 inflammasome, which in turn activates a pro-caspase-1 inducing the maturation of IL-1β and other NF-κB-dependent proinflammatory cytokines [35,38]. Among other things, IL-1β further activates the NF-κB response, closing the vicious cycle and further enhancing the inflammatory response. It has been shown that NF-κB can also act as a protective factor against excessive activation of the NLRP3 inflammasome, by eliminating its main source of activation, i.e., damaged mitochondria. NF-κB induces the expression of p62, which triggers mitophagy induced by these damaged mitochondria [39], although this mechanism may not function properly during aging. Furthermore, several studies suggest that during aging there is intensified activation of apoptosis, with the TNF-α- and mitochondria-mediated pathways likely being the most relevant in sarcopenia [40]. Mitochondria are considered the main site for integrating apoptotic signals and can induce apoptosis through multiple pathways, including caspase-dependent and caspase-independent pathways, and their function is impaired with age [23,41].

The molecular mechanisms described above increase exponentially with aging in various organs and tissues, including skeletal muscle (Figure 1) [33]. Therefore, in this tissue, these processes may account for the loss of muscle fibers and muscle function, as well as the development of age-associated sarcopenia and frailty.

## 2. Role of Clock Genes in Skeletal Muscle

### 2.1. Circadian System Organization

If we consider it from an evolutionary perspective, all species, including mammals, must adapt their vital functions to the environment for survival. As a result of this, they undergo periodic changes every 24 h known as circadian rhythms, which adapt physiological events to the daily photoperiod or light/dark cycle. Virtually every function in a living organism changes rhythmically along the 24 h period. The underlying mechanism for this type of adaptation is called the “biological clock”, and the photoperiod is the external signal or Zeitgeber that synchronizes the clock with the duration of the day. Initially, the control of circadian rhythms was attributed to oscillations of activity within a group of approximately 20,000 neurons located in the suprachiasmatic nucleus (SCN) of the hypothalamus [42,43]. This perspective has changed, and we now know that most peripheral cells, including muscle fibers, also contain clocks that function in a similar way to the central clock, but they are influenced by other environmental signals, such as the timing of food consumption and exercise. It is believed that the central clock acts hierarchically to synchronize all peripheral clocks through the daily production of melatonin, which provides information from both the clock and the calendar [35,44], revealing the existence of muscle–brain cross talk [45,46], although little is known about the mechanism by which the central and peripheral clocks communicate.

The retina possesses specialized photosensitive ganglion cells that use melanopsin as a visual pigment. These cells project through the retinohypothalamic tract to the SCN for the light entrainment of the circadian clock [47]. The retinohypothalamic tract releases glutamate in the SCN, inducing the expression of clock genes. A secondary pathway from the geniculohypothalamic tract, which primarily releases GABA, and the serotoninergic and noradrenergic inputs to the SCN can also modulate the excitatory signals of the retinohypothalamic tract [48,49]. The SCN, in turn, drives out two types of efferent signals: a homeostatic one, which affects the autonomic and neuroendocrine systems located in the hypothalamus, and a chronobiotic one, which follows a complex multisynaptic pathway to the pineal gland, where it is responsible for melatonin synthesis [50,51]. Using this mechanism, the central pacemaker regulates the circadian physiology and synchronizes the peripheral clocks through the daily production of melatonin, which also feeds back to the central clock [52]. The production of this indoleamine in the pineal gland follows a circadian rhythm, peaking at night, when the expression of the key enzymes in melatonin synthesis, arylalkylamine N-acetyltransferase (AANAT) and N-acetylserotonin O-methyltransferase (ASMT), is activated. Melatonin is not stored in the pineal; once synthetized, it is rapidly released into the bloodstream and cerebrospinal fluid, reaching all cells in the body. During the day, its synthesis is inhibited by a 460–480 nm wavelength component of the light [44].

From a molecular perspective, four main gene families (and their respective proteins), including circadian locomotor output cycles protein kaput (*Clock* (CLOCK)), brain and muscle aryl hydrocarbon receptor nuclear translocator-like 1 (*Bmal1* (BMAL1)), period genes (*Per1/2/3* (PER1/2/3)), and cryptochrome genes (*Cry1/2* (CRY1/2)), constitute the core of the biological clock. Additionally, retinoid-related orphan receptors (*Rorα/β/γ* (RORα/β/γ)) and nuclear receptor subfamily 1, group D, member 1/2 (*Rev-erbα/β* (REV-ERBα/β)) act as positive and negative modulators [53]. It has also been identified that *Chrono* and its product, CHRONO, are fundamental components of the circadian clock [54]. These genes and their products operate in transcriptional/translational feedback loops. BMAL1 interacts with CLOCK to form BMAL1:CLOCK dimers, which bind to over 6000 sites on chromatin, corresponding to approximately 3000 genes. Among them, BMAL1:CLOCK increases the expression of *Cry* and *Per* genes. CRY and PER proteins form homo- and hetero-dimers in the cytosol and then enter the nucleus, where they bind to and inactivate the BMAL1:CLOCK dimer, reducing the expression of the former. In this scenario, a further reduction in relative levels of CRY and PER does not inhibit BMAL1:CLOCK, initiating a new cycle of the clock. Besides this feedback loop, secondary loops are established, including those of ROR and REV-ERB whose expression is also controlled by the CLOCK:BMAL1 dimer. Both modulators compete to bind to the *Bmal1* promoter and the ROR response element (RORE), and they have opposing functions, with ROR activating the transcription of *Bmal1*, while REV-ERB inhibits it [55]. This mechanism oscillates for a period of 24 h. In this loop, ROR and REV-ERB act to enhance and repress the transcription of BMAL1, respectively (Figure 2) [35,53]. Recent studies have verified the existence of additional loops in the circadian clock. These autonomous loops involve the basic helix-loop-helix family members e40 and e41 (DEC1 and DEC2), which compete with CLOCK:BMAL1 for the E-box site, leading to the inhibition of their activity, as well as that of PER and CRY [56,57]. Beyond these mechanisms, the circadian clock regulates other genes that contribute to the intricate interplay of feedback loops. Thus, CLOCK:BMAL1 is capable of controlling gene expression by activating the transcription of the D-box binding protein (DBP) through binding to E-box elements in its promoter. DBP rhythmically activates the transcription of genes containing D-box elements in their promoter regions. A nuclear factor called interleukin 3 (NFIL3), whose activity is controlled by a RORE promoter region, suppresses DBP-dependent activation by competing for the D-box [58]. In turn, components of the molecular clock can be regulated through post-translational modifications, including phosphorylation by casein kinase 1ε (CK1ε), mitogen-activated protein kinase (MAPK), and glycogen synthase kinase 3β (GSK-3β), ubiquitination, and acetylation/deacetylation of histones and clock proteins [59]. The molecular mechanism and these modifications must be regulated for proper functioning, but there are numerous conditions such as aging, inflammation, or various diseases that disrupt the biological clock. The foundation of the molecular machinery of the clock was unveiled by the work of Jeffrey C. Hall, Michael Rosbash, and Michael W. Young, who were awarded the 2017 Nobel Prize in Medicine and Physiology for their contributions [60,61,62,63,64,65,66].

### 2.2. Connection between Clock Genes and Inflammation in Aging

The drastic changes in lifestyle, with increased nighttime activity and fewer hours of sleep, along with the development of chronic diseases in humans, constitute a growing threat to the proper organization of the biological clock [67]. The term “chronodisruption” (CD) is defined as a disturbance in the organization of the circadian system that alters biological rhythms and, consequently, the physiology, metabolism, and behavior of living organisms [68]. Therefore, CD is a factor that may predispose an individual to the development of multiple diseases such as cardiovascular diseases, metabolic disorders, cancer, neurodegenerative diseases like Parkinson’s or Alzheimer’s, and accelerated aging [69]. Interestingly, this relationship appears to be bidirectional, with some disease processes also associated with altered circadian function being an early symptom of the disorder [67].

In a recent study involving mice from three different age groups, it was reported that aging reduces the number of genes expressed rhythmically in a number of tissues including skeletal muscle, indicating a weakening of the circadian control [70]. This finding adds to the growing evidence suggesting a connection between biological clock disruption and the aging process. As people age, the circadian rhythm undergoes significant changes, likely due to the alteration of some components of the biological clock and the natural decline of melatonin with age, which could accelerate the aging process. For this reason, it can be said that CD promotes aging, and aging facilitates CD [69,71]. Similarly, the activity of the innate immunity is under the control of the biological clock, and its alteration is reflected in inflammatory pathologies, tissue damage, and muscle-decay pathways. This is accompanied by altered cytokine release. The disturbed muscle clock also impairs the regulation of key genes (*mTOR*, *Atrogin*, *MyoD1*, *Pgc1*, etc.), as we mentioned before, directly affecting muscle mass control [27]. Clock proteins such as BMAL1, CLOCK, PER, CRY, and the modulators RORα and REV-ERBα, play a significant role in cellular immunity, defense, and inflammation [72], and their expression is altered in aging, mainly in peripheral tissues [70,73]. In particular, BMAL1 is an essential component that links the molecular clock with immunity, thereby limiting inflammation. It performs this function in several ways. First, BMAL1 binds to CLOCK, preventing the acetylation of the p65 subunit and the activation of NF-κB by the latter. This results in a lower induction of specific genes, including cytokines and regulators of survival and proliferation, leading to a decrease in inflammation [74]. Nguyen et al. demonstrated that BMAL1 directly reduces the expression of the chemokine CCL2, thus decreasing the number of inflammatory monocytes in both blood and affected tissues [75]. Furthermore, BMAL1 regulates the circadian production of the protein nicotinamide phosphoribosyltransferase (NAMPT), which is the limiting enzyme in NAD^+^ synthesis and a cofactor of two deacetylase sirtuins, SIRT1 and SIRT3. The former inactivates NF-κB, thanks to its deacetylase activity, controlling the immune response, and playing a role in mitochondrial biogenesis and dynamics, while SIRT3 improves mitochondrial function, reducing the generation of free radicals, thereby decreasing NLRP3 inflammasome activation [35,76,77,78,79]. SIRT1, in turn, regulates various genes associated with the circadian clock, thus influencing inflammation. It has been demonstrated that SIRT1 directly associates with the BMAL1:CLOCK complex, where it can produce modifications in both proteins, and also promotes the deacetylation and degradation of PER2. SIRT1 is also necessary for the transcription of several major clock genes, including *Bmal1*, *Rorγ*, *Per2*, and *Cry1* [80,81]. It is important to mention that BMAL1 induces the expression of RORα and REV-ERBα, which activate and inhibit BMAL1, respectively, thus influencing immunity. Additionally, studies conducted in human primary smooth muscle cells and macrophages have shown that RORα induces the transcription of inhibitor of κB (IκB), preventing the translocation of NF-κB into the nucleus [82], and REV-ERBα regulates the production and release of the proinflammatory cytokine IL-6 [83]. Finally, the proteins PER and CRY have different roles in the inflammatory process. Although there are three PER proteins, it seems that PER2 is the most significant in the control of the immune system. PER2 can contribute to inflammation by restricting the activity of the BMAL1:CLOCK complex and increasing the production of INF-γ and IL-1β [84]. PER2 can also inhibit the activity of REV-ERBα, thus having a more complex role in immunity [85]. CRY1 and CRY2 inhibit different proinflammatory cytokines. The absence of these cryptochromes triggers the production of inducible nitric oxide synthase (iNOS), IL-6, and TNF-α, reflecting a proinflammatory condition, which in turn leads to increased phosphorylation of p65 and activation of the NF-κB pathway [86].

Together, these loops explain how clock genes directed by BMAL1 influence the control of innate immunity [87], promoting an anti-inflammatory state, which declines with aging, as we demonstrated in previous studies [78,88,89]. Disruption of the circadian system with age, therefore, could trigger inflammatory processes, bringing along oxidative stress, mitochondrial damage, and apoptosis mechanisms, which in skeletal muscle precede sarcopenia. In turn, clock genes appear as emerging targets against aging [90].

### 2.3. Clock Genes in Skeletal Muscle

The molecular clock is present in various cells and tissues, including muscle. Reeds et al. observed changes in protein synthesis and degradation in the skeletal muscle of rats over a 24 h period, indicating the presence of a circadian rhythm in this tissue [91]. Subsequent transcriptomic studies have identified over 200 genes that exhibit rhythmic expression in mouse skeletal muscle, including some belonging to the central molecular clock such as *Bmal1*, *Per2*, and *Cry1*, as well as specific skeletal muscle genes like *Myod1*, *Ucp3*, *Fbxo32/Atrogin1*, and *Myh1* [92,93]. Interestingly, recent studies on the slow soleus, composed of approximately 90% type 1 slow and 2A fibers, and the fast tibialis anterior, composed of about 90% fast 2B and 2X fibers, have revealed a distinct circadian gene-expression pattern in these two skeletal muscles. The findings indicated approximately two–three times more cyclically regulated genes in the soleus compared to the tibialis anterior [94].

A growing body of research supports the importance of clock genes in muscle health. Mutant mice for various clock genes have shown a decrease in muscle function. Specifically, mice homozygous for the *Clock* gene, particularly those with the truncated CLOCK protein (Δ19), exhibited a significant reduction in the expression of genes related to the sarcomere and mitochondria. This resulted in severe disruptions in their locomotor activity rhythm, muscle strength, and muscle structure. Additionally, they showed a decrease in mitochondrial content and a shorter lifespan [92,95,96,97]. On the other hand, *Clock* knockout mice (*Clock*^−/−^) had minimal muscle alterations and only a slight reduction in the duration of the circadian locomotor activity cycle. However, they experienced a decrease in lifespan and suffered from other conditions such as cataracts and a higher risk of dermatitis [98,99,100]. These differences are possibly due to the distinct mutations in the *Clock* gene between the two types of mice, with potential compensation by the NPAS2 protein in *Clock*^−/−^ mice in the absence of CLOCK. Models involving inhibitors of the molecular clock, *Per* and *Cry*, show less-significant impact on muscle health, manifesting only slight alterations in the circadian rhythm of locomotor activity. However, the simultaneous deletion of both genes has led to behavior completely devoid of rhythm [101,102,103]. On the other hand, it has been demonstrated that *Rev-erbα* is highly expressed in oxidative skeletal muscle, and its deficiency in muscle leads to a decrease in mitochondrial content and function, as well as an increase in autophagy [104].

The most extensively studied clock gene in relation to skeletal muscle is *Bmal1*. This gene plays a crucial role in muscle function, as it not only governs muscle regulation and repair [105,106], but also enhances respiration by boosting oxidative capacity in this tissue [107,108]. This is attributed to *Bmal1*’s regulation of mitochondrial dynamics and activation of mitochondrial metabolism through SIRT1 and SIRT3, respectively, by regulating NAD^+^ levels. This clock gene rhythmically controls the NAMPT-NAD-SIRTs axis, regulating the transition of mitochondria from low to high oxidative phosphorylation (OXPHOS) states and vice versa [77]. Moreover, recent studies have indicated that changes in mitochondrial morphology (fusion and fission) and the generation of new mitochondria depend on a functional circadian clock, specifically through the activation of SIRT1 and other proteins by the BMAL1:CLOCK complex [79]. These findings support the existence of a daily rhythm in the oxidative capacity of skeletal muscle [109]. Additionally, *Bmal1* also contributes to bone and cartilage development, as well as collagen synthesis and release in chondrocytes [110,111]. Various animal models have been employed in the study of the *Bmal1* gene, including *Bmal1*^−/−^ mice lacking the gene entirely, and mice with specific *Bmal1* deficiency in skeletal muscle. The former exhibit a more severe form of pathology, likely due to the involvement of multiple functions in the body in addition to muscle. This includes a significant reduction in lifespan, the development of premature sarcopenia, behavioral arrhythmia, impaired insulin sensitivity and glucose tolerance, ectopic calcification, and sterility. They also display weakness, mitochondrial dysfunction, and alterations in myofilament architecture, indicating a substantial impact on the structure and function of skeletal muscle [95,112,113,114]. To grasp the precise role of *Bmal1* in skeletal muscle, several studies have been conducted in mice deficient in *Bmal1* in skeletal muscle, including a recent study from our research group. These suggest that the *Bmal1* gene is essential for preserving muscle function and structure and for mitochondrial maintenance, thus delaying frailty and sarcopenia [111,115,116]. Other researchers identified alterations in glucose metabolism and a decrease in GLUT4 levels in the muscle tissue of these mice [117].

All of these data clearly indicates that clock genes, especially *Bmal1*, play a crucial role in skeletal muscle health, particularly in the functioning of mitochondria in this tissue.

## 3. Melatonin as a Potential Therapeutic Approach in Sarcopenia

### 3.1. Synthesis, Metabolism, and Targets of Melatonin

Melatonin, chemically known as N-acetyl-5-methoxytryptamine, is an indoleamine derived from tryptophan. It was first isolated from bovine pineal glands in 1958 by Aaron Lerner, who identified its chemical structure [118]. It is a highly conserved molecule throughout evolution, found in nearly all organisms, from bacteria and plants to invertebrates and mammals [119].

Initially, melatonin was thought to be exclusively produced by the pineal gland and was primarily investigated for its role in regulating circadian rhythms. But melatonin is also synthesized in most tissues and organs of the body, including skeletal muscle, through the same enzymatic processes as in the pineal gland. Therefore, we differentiate between two types of melatonin: pineal and extrapineal, each with distinct properties, with the latter displaying concentrations typically one or two orders of magnitude higher than pineal melatonin, depending on the tissue. This leads to significantly elevated percentage levels [120,121]. In vertebrate animals, pineal melatonin synthesis occurs during the night, in synchrony with the light–dark cycle and is controlled by the central biological clock in the SCN, as explained earlier. At the cellular level, melatonin synthesis primarily takes place in the mitochondria. It begins with the absorption of tryptophan by pinealocytes from the bloodstream, which is hydroxylated by the enzyme tryptophan hydroxylase (TPH) at position 5, giving rise to 5-hydroxytryptophan. Subsequently, 5-hydroxytryptophan is decarboxylated to form serotonin (also known as 5-hydroxytryptamine) through the action of the enzyme L-amino acid aromatic decarboxylase (AADC). Serotonin is acetylated by the enzyme AANAT, resulting in N-acetylserotonin. Then, N-acetylserotonin is methylated by ASMT, ultimately producing the melatonin molecule [122] (Figure 3). The enzymes AANAT and ASMT are crucial in melatonin synthesis, although there is controversy about which of the two acts as the limiting enzyme in the process [123]. It is important to note that pineal melatonin is not stored within the gland but is released directly into the bloodstream, from where it is distributed throughout the body. In contrast, although the same enzymes are involved in melatonin synthesis in extrapineal tissues, its production does not follow a circadian rhythm. Instead, each tissue produces the necessary amount at any given time, and melatonin remains inside the cell [120,124].

Circulating pineal melatonin has a half-life of approximately 30 min in blood and is found free or conjugated with albumin. It is primarily metabolized by the liver, resulting in 6-hydroxymelatonin and, to a lesser extent, N-acetylserotonin. Here, various isoforms of cytochrome P450, specifically enzymes CYP1A1, CYP1A2, CYP1B1, and CYP2C19, produce these metabolites [125]. Then, 6-hydroxymelatonin and N-acetylserotonin are metabolized by sulfotransferases before being excreted in urine [126]. In all tissues, melatonin can be broken down through enzymatic and non-enzymatic processes, producing the metabolites N1-acetyl-N2-formyl-5-methoxykynuramine (AFMK) and N1-acetyl-5-methoxykynuramine (AMK). These metabolites play a crucial role in the elimination of free radicals (Figure 3) [127].

To carry out its multiple functions, melatonin can interact with various receptors or cytosolic proteins, or act independently by itself. 

Although it was initially believed that melatonin could cross all biological membranes due to its lipophilic nature [128], it is actually an amphipathic molecule, with hydrophilic and hydrophobic ends. This limits its diffusion capacity through biological membranes. Research in our laboratory has shown that melatonin does not easily cross cell membranes. Instead, it accumulates on the surface of these membranes, and only a small fraction manages to penetrate into the cellular interior, saturating its intranuclear and intramitochondrial levels at certain concentrations [124]. This mechanism is essential to prevent an excess of melatonin inside the cell, as its antioxidant action could interfere with normal energy metabolism.

Melatonin receptors are widely distributed among different tissues in the body, including skeletal muscle. Two types of G-protein-coupled membrane receptors for melatonin have been described in vertebrate animals, MT1 (Mel1_a_) and MT2 (Mel1_b_). Both have seven transmembrane domains, and, upon melatonin binding, they can modulate the activity of adenylate cyclase, phospholipases C and A2, potassium and calcium channels, and guanylate cyclase through different mechanisms [129,130]. At one point, the existence of a third melatonin binding site on the cell membrane (MT3 receptor) was theorized [131]. However, it was discovered that this biological target of melatonin was actually the cytosolic enzyme quinone reductase 2 (NQO2) [132]. Other cytosolic proteins, such as calmodulin, calreticulin, tubulin, and protein kinase C, involved in calcium metabolism and cytoskeletal structure modulation, have been shown to be targets of melatonin, exerting various effects upon binding [133,134,135,136].

Melatonin also binds to nuclear receptors, specifically to the subfamily of orphan receptors related to retinoids, which include RORα, RORβ, and RORγ. These receptors are located in the cell nucleus and consist of an N-terminal domain, a DNA binding domain, and a C-terminal domain where they bind to their ligands [137]. Numerous studies, including some from our group, demonstrated the presence of melatonin in the cell nucleus, as well as its ability to bind to these receptors, particularly to RORα, regulating gene transcription and carrying out its effects [138,139,140,141,142,143]. Furthermore, like membrane receptors, nuclear receptors exhibit a circadian rhythm similar to that of melatonin, with a peak of production at 3 a.m., suggesting that their expression is regulated by melatonin levels in the blood [144].

Melatonin, along with its metabolites AFMK and AMK, is capable of directly scavenging ROS and RNS [145] (see below).

### 3.2. Actions of Pineal and Extrapineal Melatonin

Melatonin is a versatile molecule with the ability to perform multiple functions. 

On one hand, pineal melatonin released into circulation reaches every cell in the body and, upon binding to MT1 and MT2 membrane receptors, activates various pathways involved in chronobiological regulation, thus controlling peripheral clocks in different tissues, including skeletal muscle [44,120]. Here, melatonin, through clock genes, jointly coordinates various metabolic, mitochondrial, inflammatory, and structural processes in the muscle following a circadian pattern, as described earlier.

There is evidence that the degradation of clock proteins such as BMAL1, PER, CRY, and REV-ERB, as well as possibly other components, is mediated by the ubiquitin-proteasome system [146,147]. It has been suggested that melatonin may regulate this degradation by inhibiting the proteasome, as it shares similarities with its inhibitor [52,148]. Additionally, it has been observed that melatonin binds to Ca^2+^/calmodulin [149] and blocks the activity of the Ca^2+^/calmodulin II-dependent protein kinase (CaMKII) [150], which controls proteasome phosphorylation [151]. This mechanism could be how melatonin regulates clock protein degradation, synchronizing peripheral clocks and providing stability to the rhythm. However, it is currently unknown whether clock genes also control melatonin synthesis in extrapineal tissues [120]. Not in the same way as the in the pineal gland, where it has been demonstrated that the BMAL1:CLOCK complex activates the transcription of AANAT by binding to the E-box region of this gene [152], stimulating rhythmic melatonin production.

While pineal melatonin serves chronobiological functions, extrapineal melatonin acts as an antioxidant and anti-inflammatory agent, with the mitochondria being its primary target [153]. This organelle is clearly affected in aged skeletal muscle.

The antioxidant capacity of melatonin is directly linked to its ability to be produced in various cells of the body in significantly higher quantities than the concentrations of pineal melatonin found in the blood [124], exerting a local effect. Non-circadian expression of genes *Aanat* and *Asmt* has been detected in peripheral tissues, such as the heart, liver, stomach, intestines, skeletal muscle, testicles, and ovaries, among others [120,121]. Both melatonin and its metabolites, AFMK and AMK, have the ability to directly neutralize free radicals, interacting with various reactive oxygen and nitrogen species, including hydroxyl radicals (OH^•^), superoxide radicals (O2^−•^), hydrogen peroxide (H_2_O_2_), nitric oxide (NO), peroxynitrites (ONOO^−^), and peroxyl radicals (LOO^•^) [154,155]. Additionally, it has been demonstrated that this hormone is capable of indirectly eliminating free radicals by regulating the activity and expression of other antioxidant systems, an effect that could be mediated by its interaction with calmodulin and its nuclear receptor [156], or with its MT1 and MT2 membrane receptors in the case of pineal melatonin [157]. Firstly, melatonin enhances the activity of antioxidant enzymes such as glutathione peroxidase (GPx) and glutathione reductase (GRd), driving the glutathione cycle and maintaining the balance between oxidized and reduced glutathione (GSSG/GSH) [158,159]. It also stimulates γ-glutamylcysteine synthase, increasing GSH production [160], and glucose-6-phosphate dehydrogenase (G6PD), which provides the necessary NADPH for GRd [161]. Furthermore, melatonin reinforces the activity and expression of other antioxidant enzymes like superoxide dismutase (SOD) and catalase [153,162]. Since the mitochondria are the main source of free radicals, this organelle is its primary target [124]. Beyond its antioxidant function, melatonin plays a crucial role in maintaining mitochondrial homeostasis, preserving the integrity and functionality of membranes, and enhancing mitochondrial bioenergetics [145,163,164,165,166]. These functions of melatonin are especially important in muscle tissue, where mitochondria are critical organelles responsible for regulating the metabolic status of skeletal muscle [167].

Melatonin can also function as an anti-inflammatory molecule by modulating pathways of innate immunity dependent on NF-κB and the NLRP3 inflammasome. Specifically, melatonin binds to its nuclear receptor RORα, activating SIRT1, which in turn deacetylates NF-κB, inhibiting its binding to DNA and its activation, resulting in reduced expression of iNOS and pro-inflammatory cytokines such as TNF-α and IL-6, among others. Thus, melatonin prevents the activation of the NLRP3 inflammasome and caspase-1, as well as IL-1β [35,38]. Additionally, melatonin inhibits the expression of cyclooxygenase-2 (COX-2), preventing excessive production of inflammatory mediators [168]. Furthermore, in stressful situations, melatonin enhances the expression of NRF2, a transcriptional regulator of antioxidant enzymes implicated in maintaining mitochondrial homeostasis [169,170], helping in the anti-inflammatory response. Finally, melatonin also plays a crucial role in apoptosis by inhibiting it through the regulation of the BAX/BCL2 balance and the reduction of caspase-3 activity and expression [171].

All these properties of melatonin make it a potential therapeutic agent against a wide range of age-related diseases, such as sarcopenia, where disruptions in circadian rhythms, chronic inflammation, oxidative stress, mitochondrial damage, and muscle mass loss in aged muscle can be counteracted by melatonin (Figure 4).

### 3.3. Melatonin as a Link between Clock Genes and Mitochondria in Sarcopenia

Mitochondria serve as both a source of melatonin synthesis and a target for this indoleamine [124,153]. The decrease in melatonin in tissues associated with aging [172,173] appears to be linked to mitochondrial dysfunction that occurs during this process. Additionally, pineal melatonin production also experiences a significant decline with age [173], potentially leading to reduced circadian control and disruption of clock genes in various tissues. In multiple experimental conditions, including acute and chronic inflammation, aging, and sarcopenia, melatonin consistently enhanced endogenous antioxidant defense, reduced innate immunity activation, and stimulated mitochondria [38,78,88,89,115,158,174,175,176,177,178]. Furthermore, melatonin was also shown to have the capacity to restore clock gene expression, which is altered in these pathologies, as well as in other conditions such as cancer or Parkinson’s [179,180].

Specifically, our research group has demonstrated the benefits of melatonin in aging skeletal muscle. In an initial study led by Sayed et al. [181], using C57BL/6J mice of different ages (3 months, 12 months, and 24 months), the early onset of sarcopenia at 12 months was identified. This was characterized by a decrease in locomotor activity and muscle mass, accompanied by an increase in the frailty index (FI). Additionally, changes in muscle structure and ultrastructure were observed, along with a reduction in the size and loss of type 2 fibers, as well as an increase in mitochondrial size, indicating potential alterations in mitochondrial dynamics. These changes worsened in older animals. Melatonin treatment improved both the function and structure of aged mice’s muscles, while also reducing damage to mitochondria and apoptotic nuclei in the muscle. Therefore, melatonin was suggested as a potential treatment for sarcopenia [176].

Given that sarcopenia is linked to aging, a mouse model lacking NLRP3 was employed to investigate the role of inflammation. It was determined that this inflammasome plays a role in the development and progression of sarcopenia in both skeletal and cardiac muscle. This effect was mitigated with melatonin treatment [89,175,177,178]. Additionally, the expression and circadian rhythm of clock genes were assessed in these mutant mice. Aging resulted in phase changes in *Clock*, a decrease in the amplitude of *Bmal1*, *Per2*, and *Clock*, and a loss of rhythm in *Per2* and *Rorα*. On the other hand, NLRP3 altered the acrophase of *Clock*, *Per2*, and *Rorα*. Administration of melatonin in these mice allowed for a reduction in age-associated inflammation and restoration of the rhythm of altered genes [88].

Finally, given the growing evidence regarding the importance of the circadian clock in maintaining muscle tissue during aging, particularly in relation to the *Bmal1* gene, as detailed in this review, we investigated its impact in the gastrocnemius muscle using an inducible and skeletal-muscle-specific *Bmal1* knockout mouse model (iMS-*Bmal1*^−/−^) [115]. This aimed to deepen the understanding of the fundamental and underlying mechanisms of sarcopenia, while it is important to note that a mutation in this gene is not considered a genetic model for studying this condition [182]. iMS-*Bmal1*^−/−^ mice were prone to sarcopenia. These mice experienced changes in their activity/rest rhythms and muscle function, as well as alterations in muscle structure, indicating atrophy, fibrosis, and a shift towards a more oxidative profile in muscle fibers. Additionally, a reduction in mitochondrial oxidative capacity and a decrease in the number of mitochondria, accompanied by damage to them, were observed. Melatonin, through a mechanism that does not require the presence of the *Bmal1* gene, was once again able to counteract the changes produced by the absence of this clock gene in these animals.

Overall, these data posit melatonin as a potential candidate against sarcopenia, and suggest that melatonin serves as the link between clock genes and mitochondria in skeletal muscle.

## 4. Conclusions and Future Directions

This review emphasizes the crucial role of clock genes, notably *Bmal1*, in maintaining skeletal muscle health. It illuminates the emerging evidence that chronodisruption may precede the mechanisms leading to sarcopenia, an age-related condition marked by declining muscle mass and function. Melatonin emerges as a promising therapeutic option, owing to its chronobiotic, antioxidant, and anti-inflammatory properties. Notably, melatonin exhibits no toxicity in both animal and human studies [183], rendering it a safe treatment. Its influence within the mitochondria, vital in aging muscle, further underscores its potential. Combined with resistance exercise, melatonin may present an effective strategy to delay sarcopenia onset.

Understanding clock genes and mitochondrial pathways is paramount in unraveling sarcopenia’s mechanisms, leading to the discovery of potential therapeutic targets. Investigating the interplay between chronobiology and sarcopenia onset is also crucial. Further research is needed to elucidate the synergistic effects of melatonin and resistance exercise, possibly through long-term intervention studies. Additionally, evaluating melatonin supplementation’s long-term effects, especially in aging populations, is essential in establishing its safety and efficacy as a therapeutic intervention.

## Figures and Tables

**Figure 1 biomolecules-13-01779-f001:**
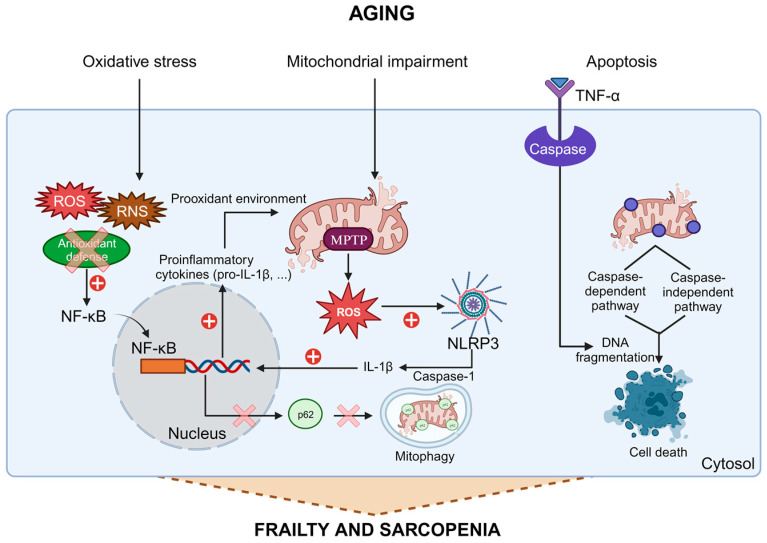
Aging, oxidative stress, mitochondrial dysfunction, and apoptosis are interconnected factors contributing to frailty and sarcopenia. “Inflammaging”, a chronic low-grade inflammation, accompanies aging, marked by increased oxidative stress, mitochondrial damage, and apoptosis. Elevated free radicals in aging activate the NF-κB pathway, releasing proinflammatory cytokines, further harming mitochondria, and activating NLRP3, and intensifying inflammation. These processes occur in various tissues, including skeletal muscle, explaining age-related muscle loss and frailty.

**Figure 2 biomolecules-13-01779-f002:**
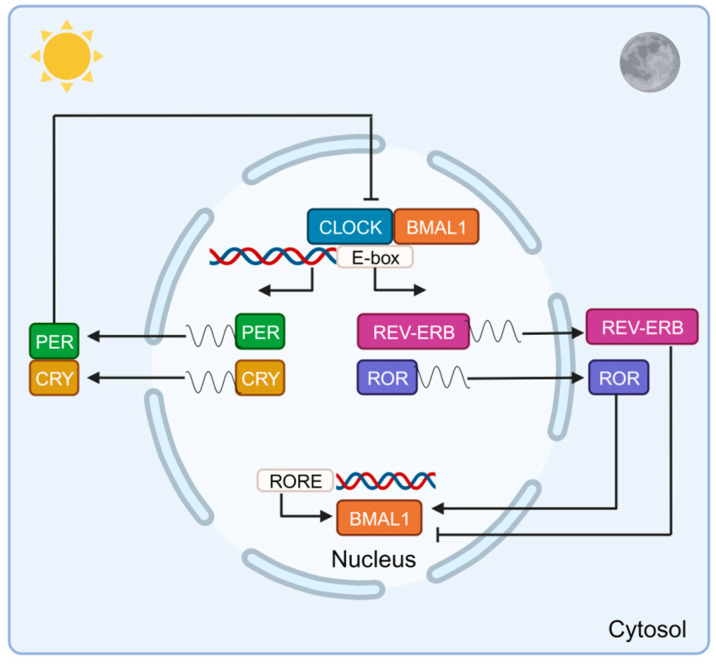
Clock genes transcriptional/translational feedback loop. The clock genes operate in 24 h cycles. BMAL1 and CLOCK form a heterodimer that activates the transcription of CRY, PER, ROR, and REV-ERB. CRY and PER combine to inhibit the activity of the BMAL1:CLOCK complex, reducing the expression of the former. The regulators ROR and REV-ERB activate and repress BMAL1, respectively.

**Figure 3 biomolecules-13-01779-f003:**
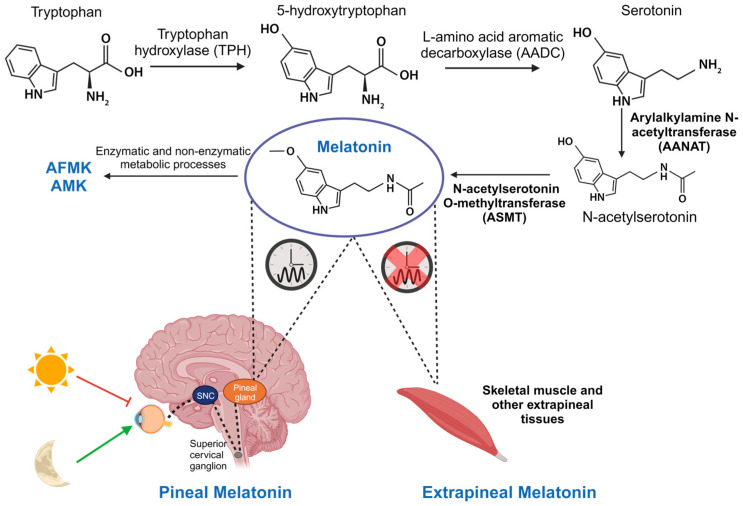
Synthesis of pineal and extrapineal melatonin. In the pineal gland, production of this molecule is nocturnal and regulated by the central biological clock in the SCN. It starts with tryptophan hydrolysis by the enzyme TPH, yielding 5-hydroxytryptophan. AADC then decarboxylates it, creating serotonin. AANAT acetylates serotonin, producing N-acetylserotonin, which is subsequently converted into melatonin by ASMT. Although the same enzymes are involved in its synthesis, in skeletal muscle and other extrapineal tissues, its production does not follow a circadian rhythm. Furthermore, in these tissues, and also in the pineal gland, melatonin is transformed into its metabolites AFMK and AMK, responsible for antioxidant functions.

**Figure 4 biomolecules-13-01779-f004:**
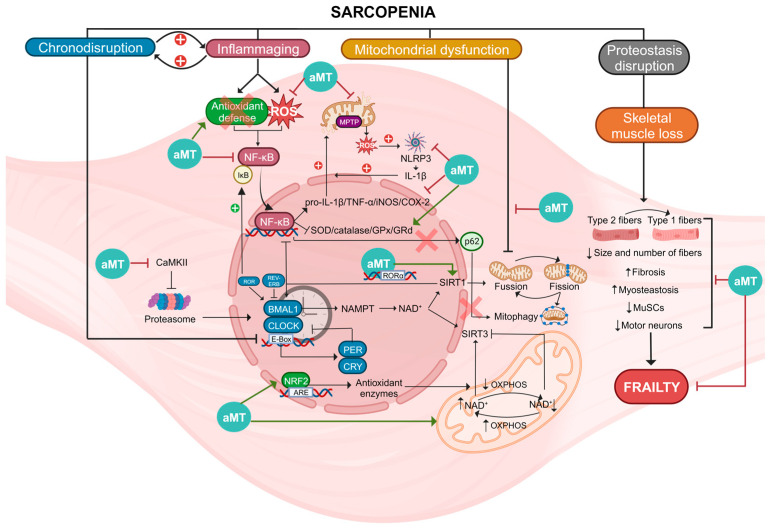
Summary of the molecular pathways involved in sarcopenia and actions of melatonin (aMT). Sarcopenia is a condition associated with several factors, including chronodisruption, inflammaging, mitochondrial dysfunction, proteostasis disruption, and skeletal muscle loss. Chronodisruption leads to the disruption of clock genes and proteins, especially BMAL1, which loses its anti-inflammatory functions. It affects the regulation of NAD^+^, which is utilized by SIRT1/3, influencing both the NF-κB pathway and OXPHOS. Additionally, the levels of NAD^+^ are influenced by these two deacetylases. Inflammaging is a process characterized by NF-κB activation caused by elevated ROS levels and diminished antioxidant defenses. This activation results in heightened levels of proinflammatory cytokines and reduced activity of antioxidant enzymes. The oxidative environment causes damage to mitochondria, leading to increased ROS and NLRP3 activation, thereby intensifying inflammation. In this scenario, p62 is unable to initiate mitophagy for the degradation of damaged mitochondria. Mitochondrial dysfunction leads to alterations in fusion, fission, and mitophagy of these organelles. Disruption of proteostasis and the loss of skeletal muscle leads to the switch of type 2 muscle fibers into type 1 and a reduction in both the size and quantity of fibers, MuSCs, and motor neurons. Moreover, there is an increase in fibrosis and myosteatosis. These mechanisms collectively disrupt skeletal muscle integrity, ultimately contributing to frailty. Melatonin, acting at diverse cellular sites, exhibits multiple advantageous properties enabling it to counteract these processes, thereby reducing muscle damage and ameliorating sarcopenia.

## Data Availability

Not applicable.

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
