# Peer review of "From Chronodisruption to Sarcopenia: The Therapeutic Potential of Melatonin"

_biomolecules, 2023, doi:10.3390/biom13121779_

Round 1
Reviewer 1 Report
Comments and Suggestions for Authors
This is an interesting review article discussing the possible application of melatonin in sarcopenia. The authors discuss efficiently aging-related processes like inflammation, oxidative stress, reduced mitochondrial capacity, and cell apoptosis in muscle cells and how melatonin interferes with them. The following comments should be considered.
1. The text on the bases of circadian clocks must be improved. Please describe the 4 secondary regulatory loops modulating master oscillation: (1) ROR (retinoid-related orphan receptor) and the nuclear receptors REV-ERB. (2) the protein ROR that also binds to the RORE element in the promotor of CLOCK, and to NFIL3 (nuclear factor, interleukin 3 regulated), thus inducing their transcription. (3) DBP (D-box binding PAR b ZIP transcription factor), a protein whose expression is controlled by BMAL1:CLOCK from the first loop which bind to the D box in the promotor region of PER. (4) DEC (AKA Basic helix-loop-helix family member e40 loop) as an ancillary circadian loop characterized by the expression of DEC and other circadian-controlled genes.
2. Minor comments. Please correct typos (indolamine, CHONO, etc) and correct fig 1 (envoronment)
Comments on the Quality of English LanguageOK
Author Response
Comments and Suggestions for Authors
This is an interesting review article discussing the possible application of melatonin in sarcopenia. The authors discuss efficiently aging-related processes like inflammation, oxidative stress, reduced mitochondrial capacity, and cell apoptosis in muscle cells and how melatonin interferes with them. The following comments should be considered.
- The text on the bases of circadian clocks must be improved. Please describe the 4 secondary regulatory loops modulating master oscillation: (1) ROR (retinoid-related orphan receptor) and the nuclear receptors REV-ERB. (2) the protein ROR that also binds to the RORE element in the promotor of CLOCK, and to NFIL3 (nuclear factor, interleukin 3 regulated), thus inducing their transcription. (3) DBP (D-box binding PAR b ZIP transcription factor), a protein whose expression is controlled by BMAL1:CLOCK from the first loop which bind to the D box in the promotor region of PER. (4) DEC (AKA Basic helix-loop-helix family member e40 loop) as an ancillary circadian loop characterized by the expression of DEC and other circadian-controlled genes.
Thank you for your comments. We have considered them and made improvements to this section accordingly (lines 277-294). We have also incorporated four new references into the manuscript as suggested.
- Minor comments. Please correct typos (indolamine, CHONO, etc) and correct fig 1 (envoronment)
We corrected typos in the manuscript and Figure 1. We changed "indolamine" to "indoleamine" (lines 255 and 616), "CHONO" to "CHRONO" (line 269), and "envoronment" to "environment" (Figure 1).
Reviewer 2 Report
Comments and Suggestions for Authors
“From chronodisruption to sarcopenia: the therapeutic potential of melatonin”
In this review paper authors characterized sarcopenia, illness being recently most often recognized in elderly people. Sarcopenia decreases body muscle mass, affecting mobility and individual independence, and becomes serious problem because of growing percentage of older population in many countries. Authors described symptoms, pathophysiology and management of sarcopenia, and presented molecular mechanisms involved in muscle’s aging. The role of clock genes in skeletal muscles and their connection to the inflammatory processes in aging was discussed, with focus on disruption of clock genes and proteins, together with loss of theirs anti-inflammatory properties. Subsequent chapter is dedicated to melatonin, its biosynthesis in pineal gland and in the extrapineal tissues, its receptors and melatonin’s role in the chronobiological regulation of tissues including skeletal muscles . Authors presented the mechanisms involved in anti-inflammatory and antioxidant effects of melatonin and indicated that melatonin has the ability to restore clock gene expression in the muscles. Additionally experimental studies revealed that melatonin treatment of aged mice improved function and structure of their muscles, reduced aged-related inflammation and restored rhythm of altered genes. Authors concluded that melatonin could be a potential treatment for sarcopenia.
Comments:
· Fig 3 This figure shows melatonin precursors and synthesis of pineal melatonin . The main source of melatonin is not pineal gland, but rather gastrointestinal tract. Melatonin is synthetized in almost all tissues of organism, including skin and skeletal muscles. Thus these tissues should be also shown on this figure as the relation between melatonin and skeletal muscles is the subject of this paper.
· The metabolites of melatonin such as AFMK and AMK are potent antioxidants, and are part of very important antioxidant cascade of melatonin (Int. J. Mol.Sci 2017). These metabolites should be presented on Fig 3.
· The list of abbreviations used in the article is recommended.
· References – high number of self-citations, which are about 20% of position from reference list .
Author Response
Comments and Suggestions for Authors
“From chronodisruption to sarcopenia: the therapeutic potential of melatonin”
In this review paper authors characterized sarcopenia, illness being recently most often recognized in elderly people. Sarcopenia decreases body muscle mass, affecting mobility and individual independence, and becomes serious problem because of growing percentage of older population in many countries. Authors described symptoms, pathophysiology and management of sarcopenia, and presented molecular mechanisms involved in muscle’s aging. The role of clock genes in skeletal muscles and their connection to the inflammatory processes in aging was discussed, with focus on disruption of clock genes and proteins, together with loss of theirs anti-inflammatory properties. Subsequent chapter is dedicated to melatonin, its biosynthesis in pineal gland and in the extrapineal tissues, its receptors and melatonin’s role in the chronobiological regulation of tissues including skeletal muscles . Authors presented the mechanisms involved in anti-inflammatory and antioxidant effects of melatonin and indicated that melatonin has the ability to restore clock gene expression in the muscles. Additionally experimental studies revealed that melatonin treatment of aged mice improved function and structure of their muscles, reduced aged-related inflammation and restored rhythm of altered genes. Authors concluded that melatonin could be a potential treatment for sarcopenia.
Comments:
Fig 3 This figure shows melatonin precursors and synthesis of pineal melatonin . The main source of melatonin is not pineal gland, but rather gastrointestinal tract. Melatonin is synthetized in almost all tissues of organism, including skin and skeletal muscles. Thus these tissues should be also shown on this figure as the relation between melatonin and skeletal muscles is the subject of this paper.
The metabolites of melatonin such as AFMK and AMK are potent antioxidants, and are part of very important antioxidant cascade of melatonin (Int. J. Mol.Sci 2017). These metabolites should be presented on Fig 3.
Thank you very much for the comments; they greatly assist us in improving the final version of the review. In response to the first two comments, we have included the new image that incorporates extrapineal melatonin production and melatonin metabolites AFMK and AMK. Additionally, we have updated the Figure 3 legend accordingly (lines 475-482) and also the main text (line 489) accordingly.
The list of abbreviations used in the article is recommended.
We have considered this and added an abbreviation list at the end of the manuscript, just before the references section (lines 694-714).
References – high number of self-citations, which are about 20% of position from reference list .
In this revision we included 6 new external citations to delve deeper into specific manuscript sections, reducing the percentage of self-citations. However, it's expected to observe a certain number of citations from our group, given our extensive work on melatonin and its properties over decades.
Reviewer 3 Report
Comments and Suggestions for Authors
J. Fernandez-Martinez et al. addresses the role of circadian rhythm and melatonin in the complex pathology of sarcopenia. It is an interesting topic and the manuscript is well written, but the organisation of the manuscript should be improved to make it more focussed and logic. Sarcopenia is widely studied and many reviews exist, but melatonin is often not discussed. To my opinion, part of the information in the first part of the review (1. Sarcopenia: age-related loss of muscle mass and function), is not needed and distracts from the topic, also many references of other reviews were used in this section. In contrast, some processes that are discussed in later sections are missing and should be added, or the order in manuscript should be changed. In addition, this review should clearly indicate the limitations/boundaries, rather than touching upon many aspects of sarcopenia very generally.
Specific questions/suggestions:
#49 “Clearly, aging is becoming one of the main concerns of humanity.” Consider rephrasing as aging itself is not the problem, but age-related diseases and other associated challenges.
#58 “…, coupled with a sustained decrease in fertility levels, …” where in reference 3 is this discussed?
#146 “..(organelles, cytoplasmic content).” Please specify more, which components are most relevant?
#165 – 172 states that additional mechanisms contribute to this functional decline in muscle strength in addition to muscle mass loss. As phrased now, this implies to me that these mechanisms act independently from the muscle mass decline. If this is meant, please elaborate on this and add references of research papers that demonstrated this.
#184 “Free radicals increase” please specify if you mean increased production or increase availability due to reduced scavenging
#184 – 193 please rephrase to make clear that mitochondria is main source of ROS formation, balance with NFkb, and place references after sentence for which reference was used, rather than combining them in line 193.
#199-202 specify type of apoptosis, also in figure 1.
Figure 1. I understand that not all connections and elements are shown for clarity, but this is a bit too simplified picture to my opinion. Also “Frailty and sarcopenia” in figure is too general. These processes contribute to reduced cell functioning and/or cell death, but do not explain age-related muscle loss and frailty. This figure could be deleted to my opinion.
#248 the “because” in this sentence is unclear to me.
#252 “four main genes”, are rather four gene families. Also, always write full gene name at first use.
#266 elaborate on the opposite effects of ROR and REV-ERB
Figure 2. Nice figure, would be very helpful if also expression pattern of key gene during 24h would be shown or if day/night could be indicated in figure 2.
#288 I suggest to add “may” as not fully certain
#295 rephrase weakened
#297-299 repetition of a few sentences earlier.
#304 the key genes listed here and associated processes should be discussed in section 1 to my opinion.
#307 “…their genes” add target?
#311 consider rephrasing “…. Reduction of the induction…”
SIRT1 also has a role in mitochondrial biogenesis
2.3 What is effect on different muscle fiber types?
#418 which percentage of melatonin is pineal?
#422 please add which steps do occur in the mitochondria, and which not
Figure 4: very nice figure!
3.3 would be helpful to have a bit more specific data rather than just …. Was increased or … decreased.
Author Response
Comments and Suggestions for Authors
J. Fernandez-Martinez et al. addresses the role of circadian rhythm and melatonin in the complex pathology of sarcopenia. It is an interesting topic and the manuscript is well written, but the organisation of the manuscript should be improved to make it more focussed and logic. Sarcopenia is widely studied and many reviews exist, but melatonin is often not discussed. To my opinion, part of the information in the first part of the review (1. Sarcopenia: age-related loss of muscle mass and function), is not needed and distracts from the topic, also many references of other reviews were used in this section. In contrast, some processes that are discussed in later sections are missing and should be added, or the order in manuscript should be changed. In addition, this review should clearly indicate the limitations/boundaries, rather than touching upon many aspects of sarcopenia very generally.
Thank your very much for your comments. We feel that this first part of the review should be maintained because Biomolecules is not a specific journal devoted to sarcopenia, but it is mor multidisciplinary one. So, the readers of this journal in general, and the special issue particularly are not only experts on sarcopenia and a brief vision of the pathophysiology of sarcopenia would be very welcome for them.
Specific questions/suggestions:
#49 “Clearly, aging is becoming one of the main concerns of humanity.” Consider rephrasing as aging itself is not the problem, but age-related diseases and other associated challenges.
We took it into consideration and replaced "Clearly, aging is becoming one of the main concerns of humanity" for "Clearly, the aging of the population, along with the emergence of age-related diseases and other associated challenges, is becoming a primary concern for humanity" (now, lines 49-51).
#58 “…, coupled with a sustained decrease in fertility levels, …” where in reference 3 is this discussed?
These comments stem from our interpretation of the information and graphs sourced from the latest report by the United Nations (United Nations, DESA, Population Division. World Population Prospects 2022. http://population.un.org/wpp/), which we discussed based on the information in reference 3. However, we also include the first reference for greater clarity (now, lines 59-61).
#146 “..(organelles, cytoplasmic content).” Please specify more, which components are most relevant?
We provided further details. We specified the most relevant organelles and proteins (now, lines 145-153). We have also adjusted the reference section accordingly.
#165 – 172 states that additional mechanisms contribute to this functional decline in muscle strength in addition to muscle mass loss. As phrased now, this implies to me that these mechanisms act independently from the muscle mass decline. If this is meant, please elaborate on this and add references of research papers that demonstrated this.
We agree with this comment. The way it is written may seem confusing because it is not about independent processes. We have rephrased it in the manuscript accordingly (now, lines 172-176).
#184 “Free radicals increase” please specify if you mean increased production or increase availability due to reduced scavenging
We appreciate this comment; however, we believe that this specification would be redundant. We mentioned it a few lines above (now, lines 184-188) where we specified that it pertains to both types: increased production and increased availability due to reduced scavenging.
#184 – 193 please rephrase to make clear that mitochondria is main source of ROS formation, balance with NFkb, and place references after sentence for which reference was used, rather than combining them in line 193.
We have updated the manuscript accordingly (now, lines 195-201). We have also modified the reference section accordingly.
#199-202 specify type of apoptosis, also in figure 1.
We specified types of apoptosis in the manuscript and included a new reference (now, lines 207-211). Additionally, we modified Figure 1, and the reference section has been adjusted accordingly.
Figure 1. I understand that not all connections and elements are shown for clarity, but this is a bit too simplified picture to my opinion. Also “Frailty and sarcopenia” in figure is too general. These processes contribute to reduced cell functioning and/or cell death, but do not explain age-related muscle loss and frailty. This figure could be deleted to my opinion.
While it is a somewhat general figure, we believe it is useful, especially for readers who may not be as familiar with the topic, to connect these general processes with frailty and sarcopenia. Subsequently, we found Figure 4, where connections and elements are presented in more detail and specifically related to skeletal muscle.
#248 the “because” in this sentence is unclear to me.
We have omitted "because" and adjusted the text accordingly (now, lines 258-259).
#252 “four main genes”, are rather four gene families. Also, always write full gene name at first use.
We took it into consideration and changed "four main genes" to "four main gene families" (now, line 262). Also, we added the full gene names (now, lines 263-268).
#266 elaborate on the opposite effects of ROR and REV-ERB
We have incorporated additional details in the manuscript regarding the contrasting effects of ROR and REV-ERB (now, lines 277-281). Furthermore, we have included a new reference and updated the reference section accordingly.
Figure 2. Nice figure, would be very helpful if also expression pattern of key gene during 24h would be shown or if day/night could be indicated in figure 2.
We added information to Figure 2 to show the expression timing (day/night) of the primary clock genes in these loops.
#288 I suggest to add “may” as not fully certain
We took it into consideration and added "may" (now, line 317).
#295 rephrase weakened
We changed "weakened" to "weakening" (now, line 324).
#297-299 repetition of a few sentences earlier.
We believe that, although this sentence summarizes some previously stated sentences, it is useful as a link between the previously discussed chronodisruption and aging, emphasizing their bidirectional relationship and introducing new, pertinent information, such as the natural decline of melatonin with age. However, if this reviewer deems it appropriate, it could be omitted.
#304 the key genes listed here and associated processes should be discussed in section 1 to my opinion.
We considered it and discussed certain aspects of that point in section 1 (now, lines 145-153, also 333). We have adjusted the reference section accordingly.
#307 “…their genes” add target?
In this expression, we aimed to mention the previously discussed clock proteins. We have adjusted the text based on feedback to enhance accuracy (now, line 336).
#311 consider rephrasing “…. Reduction of the induction…”
We considered this comment and replaced "reduction of the induction" for "lower induction" (now, line 341).
SIRT1 also has a role in mitochondrial biogenesis
Certainly. In fact, we discuss the role of SIRT1 in mitochondrial biogenesis and dynamics in the following section (2.3), not in the current one (2.2), where our focus is on exploring the relationship between clock genes and inflammation. Nevertheless, we have introduced this information in section 2.2 as well to prevent any confusion (now, lines 348-349). We have also adjusted the reference section accordingly.
2.3 What is effect on different muscle fiber types?
This is an interesting question. In this section (2.3), we included information about the differential effect of clock genes on muscle tissues with varying types of fibers (fast and slow) (now, lines 385-390).
#418 which percentage of melatonin is pineal?
Now, we have addressed this question in the manuscript (now, lines 451-454).
#422 please add which steps do occur in the mitochondria, and which not
Thank you for the observation. Upon consulting the cited reference and additional literature, all sources agree that melatonin is primarily produced in the mitochondria. However, the cited reference simply mentions "Depending on the organism, not all of the events necessarily take place in the chloroplasts or mitochondria of every species", without providing further details, so we prefer not to consider it. Therefore, we choose to omit this (now, lines 457-458).
Figure 4: very nice figure!
Thank you! We appreciate this comment.
3.3 would be helpful to have a bit more specific data rather than just …. Was increased or … decreased.
We appreciate the comment, but, while it is true that it could be helpful to show specific data about the results that appear in section 3.3, we believe that the text in this section is clear, and the data can be consulted in the original articles cited, where the information is more detailed.
Round 2
Reviewer 3 Report
Comments and Suggestions for Authors
Fernández-Martínez et al. drastically improved clarity of the manuscript, no further adaptation needed.